# Identification and Characterization of Novel Fusion Genes with Potential Clinical Applications in Mexican Children with Acute Lymphoblastic Leukemia

**DOI:** 10.3390/ijms20102394

**Published:** 2019-05-15

**Authors:** Minerva Mata-Rocha, Angelica Rangel-López, Elva Jiménez-Hernández, Blanca Angélica Morales-Castillo, Carolina González-Torres, Javier Gaytan-Cervantes, Enrique Álvarez-Olmos, Juan Carlos Núñez-Enríquez, Arturo Fajardo-Gutiérrez, Jorge Alfonso Martín-Trejo, Karina Anastacia Solís-Labastida, Aurora Medina-Sansón, Janet Flores-Lujano, Omar Alejandro Sepúlveda-Robles, José Gabriel Peñaloza-González, Laura Eugenia Espinoza-Hernández, Nora Nancy Núñez-Villegas, Rosa Martha Espinosa-Elizondo, Beatriz Cortés-Herrera, José Refugio Torres-Nava, Luz Victoria Flores-Villegas, Laura Elizabeth Merino-Pasaye, Vilma Carolina Bekker-Méndez, Martha Margarita Velázquez-Aviña, María Luisa Pérez-Saldívar, Benito Alejandro Bautista-Martínez, Raquel Amador-Sánchez, Ana Itamar González-Avila, Silvia Jiménez-Morales, David Aldebarán Duarte-Rodríguez, Jessica Denisse Santillán-Juárez, Alejandra Jimena García-Velázquez, Haydeé Rosas-Vargas, Juan Manuel Mejía-Aranguré

**Affiliations:** 1CONACyT-Unidad de Investigacion Medica en Epidemiologia Clinica, Hospital de Pediatria, Centro Medico Siglo XXI, IMSS, 06720 Mexico City, Mexico; mmata@conacyt.mx (M.M.-R.); sero__82@hotmail.com (O.A.S.-R.); 2Unidad de Investigacion Medica en Genética Humana, Hospital de Pediatria, Centro Medico Nacional Siglo XXI, IMSS, 06720 Mexico City, Mexico; 3Coordinacion de Investigacion en Salud, Unidad Habilitada de Apoyo al Predictamen, Centro Medico Siglo XXI, IMSS, 06720 Mexico City, Mexico; angelica.rangell@imss.gob.mx; 4Servicio de Hematologia Pediatrica, Hospital General “Gaudencio González Garza”, Centro Medico Nacional (CMN) “La Raza”, IMSS, 02990 Mexico City, Mexico; elvajimenez@yahoo.com (E.J.-H.); laura.espinosah@imss.gob.mx (L.E.E.-H.); nanuvi_2401@yahoo.com.mx (N.N.N.-V.); 5Servicio de Oncología, Hospital Pediatrico de Moctezuma, Secretaria de Salud de la Ciudad de Mexico, Ciudad de Mexico, 15530 Mexico City, Mexico; torresoncoped@live.com.mx; 6Unidad de Investigacion Medico en Epidemiologia Clinica, Hospital de Pediatria, Centro Medico Nacional Siglo XXI, IMSS, 06720 Mexico City, Mexico; anne_tilo_kurt@hotmail.com (B.A.M.-C.); enrique20_85@hotmail.com (E.Á.-O.); jcarlos_nu@hotmail.com (J.C.N.-E.); afajardo@unam.mx (A.F.-G.); janetflores22@yahoo.com.mx (J.F.-L.); maria_luisa_2000_mx@yahoo.com (M.L.P.-S.); turunci@gmail.com (D.A.D.-R.); 7Laboratorio de Secuenciación, Division de Desarrollo de la Investigacion, Centro Medico Nacional Siglo XXI, IMSS, 06720 Mexico City, Mexico; gonzaleztorrescaro@gmail.com (C.G.-T.); javier_gc50@hotmail.com (J.G.-C.); 8Servicio de Hematologia, UMAE Hospital de Pediatria, Centro Medico Nacional Siglo XXI, IMSS, 06720 Mexico City, Mexico; jorge.martin.trejo@gmail.com (J.A.M.-T.); kas_anastacia@yahoo.com (K.A.S.-L.); aufgefuhrt@hotmail.com (B.A.B.-M.); 9Servicio de Oncología, Hospital Infantil de Mexico Federico Gómez, Secretaria de Salud, 06720 Mexico City, Mexico; auroramedina@aol.com.mx; 10Servicio de Onco-Pediatria, Hospital Juarez de Mexico, Secretaria de Salud, 07760 Mexico City, Mexico; penaloza_6@yahoo.es (J.G.P.-G.); m_mvelazquez@yahoo.com.mx (M.M.V.-A.); 11Servicio de Hematologia Pediatrica, Hospital General de Mexico, Secretaria de Salud, 06726 Mexico City, Mexico; rmespinosa1605@hotmail.com (R.M.E.-E.); beatrizcortes101087@gmail.com (B.C.-H.); 12Servicio de Hematologia Pediatrica, Centro Medico Nacional “20 de Noviembre”, ISSSTE, 03229 Mexico City, Mexico; victoriabanco@yahoo.com.mx (L.V.F.-V.); sketch0712@gmail.com (L.E.M.-P.); 13Unidad de Investigacion Medico en Inmunologia e Infectologia, Hospital de Infectologia “Dr. Daniel Méndez Hernández”, “La Raza”, IMSS, 02990 Mexico City, Mexico; bekkermendez@yahoo.com; 14Servicio de Hematologia Pediatrica, Hospital General Regional “Carlos McGregor Sanchez Navarro”, IMSS, 03100 Mexico City, Mexico; dibs_amador@hotmail.com (R.A.-S.); itamarga@hotmail.com (A.I.G.-A.); 15Laboratorio de Genomica del Cancer del Instituto Nacional de Medicina Genomica (INMEGEN), 14610 Mexico City, Mexico; sjimenez@inmegen.gob.mx; 16Servicio de Hemato-Oncologia Pediatrica, Hospital Regional 1° de Octubre, ISSSTE, 07300 Mexico City, Mexico; jessydenise22@hotmail.com (J.D.S.-J.); ale.garciavelazquez@gmail.com (A.J.G.-V.); 17Coordinación de Investigacion en Salud, IMSS, Torre Academia Nacional de Medicina, 06720 Mexico City, Mexico

**Keywords:** leukemogenesis, chromosomal translocation, fusion gene, RNA sequencing, risk stratification, genetic variations

## Abstract

Acute lymphoblastic leukemia is the most common type of childhood cancer worldwide. Mexico City has one of the highest incidences and mortality rates of this cancer. It has previously been recognized that chromosomal translocations are important in cancer etiology. Specific fusion genes have been considered as important treatment targets in childhood acute lymphoblastic leukemia (ALL). The present research aimed at the identification and characterization of novel fusion genes with potential clinical implications in Mexican children with acute lymphoblastic leukemia. The RNA-sequencing approach was used. Four fusion genes not previously reported were identified: *CREBBP-SRGAP2B*, *DNAH14-IKZF1*, *ETV6-SNUPN*, *ETV6-NUFIP1*. Although a fusion gene is not sufficient to cause leukemia, it could be involved in the pathogenesis of the disease. Notably, these new translocations were found in genes encoding for hematopoietic transcription factors which are known to play an important role in leukemogenesis and disease prognosis such as *IKZF1*, *CREBBP*, and *ETV6*. In addition, they may have an impact on the prognosis of Mexican pediatric patients with ALL, with the potential to be included in the current risk stratification schemes or used as therapeutic targets.

## 1. Introduction

Acute lymphoblastic leukemia (ALL) is the most common cancer in children under 15 years old. Mexico City and Hispanic children in general have one of the highest incidences and mortality rates of this cancer. Additionally, more than half of these children (58.8%) are classified with a high risk of relapse at the moment of diagnosis confirmation [1,2]. In Mexico City, ALL represents the second cause of death in children aged between 1 and 14 years [3], with an annual incidence of 49.5 cases per million children under 15 years old [1]. It has been reported that ALL is a disease associated with a great genetic variation; because of this, we hypothesize that the high incidence of childhood ALL in our population may be due, among other factors, to a distinctive genetic variation [4].

Several studies have realized the genomic characterization of children with ALL, finding distinct patterns of genes and pathways altered with somatic variants, deletion, chromosomal translocations, and the presence of DNA copy number alterations. These studies have determined that genomic heterogeneity in ALL is largely because each patient had a unique genome with specific genomic aberrations, which are used to reclassify ALL into subtypes [4,5]. We approach this study from the perspective of the chromosomal translocations because these have been considered as initiating events in leukemogenesis (premalignant clone) and play a central role in the malignant transformation of many cancers, including ALL. Nevertheless, they are not sufficient to originate leukemia, which requires accumulation of complementary genetic lesions [6,7]. Chromosome translocations are the product of a recombination or juxtapositioning of separate genes, which results in the dysregulation of the genes involved, oncogene activation, and coding for hybrid proteins with altered properties [8].

Until now, the advances in the molecular characterization of ALL have resulted in the identification of chromosomal translocations and fusion genes associated with ALL prognosis and targeting for therapy. For example, *BCR–ABL1* fusion or *Philadelphia* positive (*Ph*+) involves a recombination between *ABL1* and *BCR* genes at chromosome 9 and 22, respectively. It encodes for a constitutively active tyrosine kinase with transforming capacity. Its prevalence is between 3% and 5% of pediatric ALL cases and it is associated with poor prognosis [4]. Patients with this translocation receive specific treatment with tyrosine kinase inhibitors (TKIs) and their survival rates have improved over the last years in developed countries. On the other hand, *KMT2A* (*MLL*) fusions are present in 60–80% of ALL children under 1 year of age, whereas its prevalence is low (3%) in older children. They are associated with a high incidence of early relapses and low survival rates [9]. The most frequent translocation of children with ALL is *ETV6-RUNX1* (*TEL-AML1*), which is associated with the highest survival rates observed in this disease [10]. Nonetheless, it has been reported that its prevalence varies across populations, with the Mexican population having the lowest frequencies of detection despite the use of different methodologies (3.8–7.4%) [11]. This situation was one of the strongest reasons that led us to conduct the present research in order to investigate what other unknown clinically relevant translocation could be found in Mexican children with this type of cancer, which can help diagnosis, risk stratification, and targeted therapy.

Chromosomal rearrangement detection requires molecular methods such as fluorescence in situ hybridization (FISH), comparative genomic hybridization (CGH), and/or RT-PCR. With the utilization of next-generation sequencing (NGS) technologies, especially RNA sequencing (RNA-seq), it is possible to simultaneously sequence multiple DNA fragments at a high sequencing depth, increasing the capacity to detect and characterize genomic aberrations at a more detailed level [12]. With RNA-seq, novel chromosomal translocations have been discovered in hematology malignancies with more sensitivity and specificity [13]. In this study, we characterized novel fusion genes present in Mexican children and analyzed the relationship between the functions of the genes that comprise the novel fusion genes and clinical features.

## 2. Results

### 2.1. Clinical Features of Patients

The clinical data of the 27 patients included are displayed in Table 1. Of these, 23 were patients diagnosed with ALL. We included four patients with a presumptive diagnosis of leukemia; however, three were subsequently confirmed with infectious diseases and one patient with bicytopenia. Thirteen ALL patients were male and 14 were female, with a mean age of 6.8 years (range 0.6–13 years), 22 had PreB-lineage, and 1 had a T cell immunophenotype. All patients achieved complete remission after induction therapy. Follow-up information was only obtained for ALL patients (Table 1). ALL cases were treated according to chemotherapy treatment used in participating hospitals. Three patients died without relapse (199MO, 63MO, and 28MO) and only one relapsed and died (74MO). The other 19 patients maintained complete remission for the follow-up period.

### 2.2. Novel Fusion Transcripts

In this study, RNA-seq assay was conducted to identify translocations in bone marrow samples. Twelve different fusion transcripts were identified in nine samples, but two of them were not possible to amplify by PCR (*GLYRI-SLC9A8* and *WDR74-RCC1).* As expected, no fusion genes were detected in the cases without leukemia. All fusions were found to be in-frame, of which five (*E2A-PBX1*, *ETV6-RUNX1*, *BCR-ABL*, *MLL-AF4*, *EP300-ZNF384* [14]) had already been reported, and four *(CREBBP-SRGAP2B*, *DNAH14-IKZF1*, *ETV6-SNUPN*, *ETV6-NUFIP1*) (Table 1 and Table 2) were not previously reported. We validated the bioinformatic result with PCR amplification and Sanger sequencing using cDNA or the DNA of the original sample and specific primers (Appendix A).

#### 2.2.1. *CREBBP-SRGAP2B* t (16;1) (p13.3;q21.1)

*CREBBP-SRGAP2B* fusion transcript was detected in one case (74MO). Interestingly, in this case, *BCR-ABL* (minor) was also identified (Table 1). This patient was treated with targeted (minor *BCR-ABL*) molecular therapy. Notwithstanding, this patient presented isolated central nervous system relapse (CNS) at two years after diagnosis confirmation, and died two months after as a consequence of septic shock. Sequencing analysis of the RT-PCR product confirmed that exon 1 with the promoter region of *CREBBP* was fused to exons 2-1 of *SRGAP2B,* which gave rise to the loss of coding sequence (CDS) of both genes. The predicted structure is shown in Figure 1 and coverage and the reading depth are displayed in Appendix A, and the PCR with genomic DNA suggests the existence of more of than one breakpoint site (Figure 1, lane 2).

#### 2.2.2. *DNAH14-IKZF1* t (1;7) (q42.12;7p12.2)

This fusion gene was observed in a 1.8-year-old girl with standard-risk Pre-B ALL (179MO) who achieved complete remission after the induction of the remission phase. The bioinformatic analysis reported different sites of fusion between *DNAH14* (exon 29, 33, 34 and 36) and *IKZF1*(exon 4 and 5, coverage and reading depth are displayed in Appendix A), suggesting that these fusions resulted from alternative splicing. By genomic DNA-PCR-validation of the original sample, we identified the breakpoint between exon 36 of *DNAH14* and exon 4 of *IKZF1* (Figure 2), resulting in a constituted fusion for the first 36 exons of the *DNAH14* and exon 4 and upstream exons of *IKZF1*. This *DNAH14-IKZF1* was in-frame and predicted to encode a chimeric protein where the final region comprising both the four-zinc finger and the dimerization of *IKZF1* are truncated.

#### 2.2.3. *ETV6-SNUPN* t (12;15) (p13.2;q24.2) and *ETV6-NUFIP1* t (12;13) (p13.2;q14.12)

These gene rearrangements were detected in a female adolescent (28MO case) diagnosed with high-risk ALL. Two months after diagnosis confirmation and one month after complete remission achievement, she presented severe neutropenia related with chemotherapy toxicity, septic shock, pneumonia, multiple organ failure, and death. The RT-PCR-Sanger indicated that the breakpoints were located on intron 1 of *ETV6* and intron 5 of *NUFIP1* in the case of the *ETV6-NUFIP1* translocation; whereas, for *ETV6-SNUPN* gene rearrangement, the breakpoints were located on intron 2 of *ETV6* and intron 1 of *SNUPN*. Both fusions involved the first exon and the promoter region of *ETV6* (Figure 3, and coverage and reading depth are displayed in Appendix A).

#### 2.2.4. *EP300-ZNF384* t (22;12) (q13.2;p13.31;)

The fusion *EP300-ZNF384* had been previously reported in other populations and associated with the prognosis of children with PreK ALL subtype. Notably, this had not been reported in Mexican children. This fusion was identified in the 197MO case: a girl of 9.4 years diagnosed with Pre-B ALL who achieved complete remission after induction therapy (Table 1). The structure and sequences of the *EP300-ZNF384* fusion are depicted in Figure 4. Exon 6 of EP300 was fused to exon 3 of *ZNF384* in-frame (Figure 4B, and coverage and reading depth are displayed in Appendix A), and the same structure was previously reported. This fusion included the initiation codon of the *ZNF384* gene and the fusion transcripts were predicted to encode a fusion protein of 110-kDa with 1027 amino acids, with this structure containing the transcriptional adapter zinc-finger 1 (TAZ1) domain in the cysteine–histidine-rich region 1 (CH1) of EP300 and the entire ZNF384 protein.

## 3. Discussion

In this study, we identified new fusion genes that can provide prognostic information because they involve important gene transcription factors with an essential role in lymphoid differentiation: these are *CREBBP*, *IKZF1*, and *ETV6*. Additionally, it has been reported that the mutations and deletions of these genes were related to prognosis in ALL children (Table 2).

For example, CREB-binding protein (CBP) is a transcriptional coactivator with intrinsic histone acetyltransferase (HAT) activity that interacts with numerous transcriptions factors. It has an important function in cell growth, division, and differentiation [20,21]. *CREBBP* is fused with *SRGAP2B*; its gene encodes for a protein belonging to the GTPase Rho SLIT-ROBO family which has been noted as participating in the transcriptional regulation pathways of brain development [22]. *CREBBP*-*SRGAP2B* fusion only retained the first exon of *CREBBP* gene and lost important structural regions such as the C-terminal (Figure 1). The C-terminal region is necessary to interact with the basal transcription factor TFIIB, considered to be a key component of the transcriptional machinery [23]. The presence of this translocation could be associated with the relapse of Mexican ALL children, given the fact that a child with this gene rearrangement had a relapse in the CNS. In other studies, it has been reported that deletions or mutations in *CREBBP* are associated with relapse and glucocorticoid resistance [15]. For this reason, it would be very important to evaluate the impact of this novel translocation in the prognosis of children with ALL and its association with drug resistance and relapse in future studies [24]. It is worth noting that in this patient, *BCR-ABL1* (minor) was also detected. In this regard, several molecular studies have provided evidence that the most frequent mode of acquired resistance is the acquisition of point mutations. We hypothesize that the coexistence of different fusion genes could induce mechanisms of drug resistance, as has been reported for fusions in tyrosine and BRAF kinases [25].

Furthermore, in the present research, the coexistence of different fusion genes was observed in two ALL cases (8.6%). This frequency is high with respect to that reported in other populations (0.24%) [26]. It is important to take into consideration that conventional molecular methods have important limitations for the detection of multiple genetic alterations occurring in the same patient, and the NGS technology allows the identification of the coexistence of different fusions with a high sensitivity and specificity [12,13]. Using this may improve the current risk stratification and chemotherapy assignment of ALL children. On the other hand, because the genomic heterogeneity in ALL is large, different mutations, deletions, and fusion genes can be co-occurring in the same patient. In this regard, we did not analyze if the fusion genes found in the present study were co-occurring with other known mutations because it was not the main objective. This is a limitation of our study, and it will be relevant to deeply explore this association in future investigations.

To the best of our knowledge, there is no information regarding the impact of the *DNAH14-IKZF1* translocation on the prognosis of leukemia patients. Therefore, it would be relevant to know what the frequency of this translocation is in a greater number of patients and if a relationship with prognosis exists. The only patient who was positive for *DNAH14-IKZF1* was classified as at standard risk; currently, she is alive and in complete remission after 2 years of treatment initiation (179MO, Table 1). However, a longer follow-up is required to discard an association with late relapses because it has been documented that deletions in IKZF1 gene have been associated with high rates of relapse in pediatric patients with ALL [16]. In addition, there is evidence to support a plausible role of this alteration in the prognosis of a child with ALL. As has been noted, *IKZF1* is a transcription factor that belongs to the family of zinc-finger DNA-binding protein and is critical for normal hematopoiesis and lymphoid system development. Different isoforms have been described and most share a common C-terminal domain (exon8), which contains two zinc fingers necessary for dimerization and interaction with other proteins [27,28]. On the other hand, the *DNAH14-IKZF1* fusion generates a protein that interrupts the formation of the four-zinc finger encoded in exons 4 and 6 as well as the dimerization sites. The *DNAH14-IKZF1* fusion is similar to the IK6 isoform of *IKZF1* [17]. It has been mentioned that the IK6 isoforms of *IKZF1* and DNA-binding domain point mutations in mouse models of *BCR-ABL1-positive* leukemia resulted in dasatinib resistance, cellular mislocalization, and the induction of stem cells [18]. Considering this information, the *DNAH14-IKZF1* fusion could provoke a functionally inactive allele of *IKZF1* (“loss-of-function”) and it is possible that this gene rearrangement could have prognostic and therapeutic implications; however, further study is required.

Two different translocations related with the ETV6 gene were detected in the same patient (*ETV6-SNUPN* and *ETV6-NUFIP1*) (Table 1, Figure 3). *SNUPN* (Snurportin 1) is an snRNP-specific nuclear import receptor and essential for proliferation and chromosome region maintenance [29], while the *NUFIP1* gene encodes for a nuclear RNA-binding protein that contains a C2H2 zinc finger motif and a nuclear localization signal. As it has been previously described, the ETV6 gene encodes an ETS (E-twenty-six) family transcription factor which has an important role in hematopoiesis, as well as in malignant transformation when *ETV6* translocations, insertions, or inversions are present [19]. It has been pointed out that *ETV6* is a highly promiscuous gene since it has been reported as having 30 ETV6 partner genes in cancer [19]. The most commonly reported partner in ALL children is *ETV6*-*RUNX1,* which is associated with a good prognosis of the disease. However, the patient who was positive for the *ETV6-SNUPN* and *ETV6-NUFIP1* translocations died as result of chemotherapy-related toxicity two months after diagnosis. The role that *ETV6*-*SNUPN* and *ETV6*-*NUFIP1* play in the prognosis of Mexican children with ALL ought to be determined in future investigations.

*ZNF384* is a C2H2-type zinc finger protein which functions as a transcription factor, and fusion genes with this gene have been reported in ~3% of children with B-cell precursor acute lymphoblastic leukemia. Specifically, *EP300*-*ZNF384* has been associated with a relatively advanced age at diagnosis, no significant elevation of the white blood cell (WBC) count at presentation, and a favorable response to conventional chemotherapy in comparison with patients with *MLL* translocations [14,30]. These characteristics were observed in the 197MO case who was positive for the *EP300-ZNF384* fusion, 9.4 years at diagnosis and 2430 WBC 10^6^ cell/L. Additionally, the patient is alive and in complete remission after 4 years of treatment initiation. The incorporation of *EP300-ZNF384* detection in the routine diagnostic panel of translocations of Mexican children with ALL could help in achieving better risk stratification and treatment.

## 4. Materials and Methods

### 4.1. Patients and Samples

This study was conducted in accordance with the principles embodied in the Declaration of Helsinki and was approved on 24 September 2013 by El Comité de Ética en Investigación del Instituto Mexicano del Seguro Social (IMSS) with project identification code R-2013-785-068. We examined 27 bone marrow (BM) samples obtained from pediatric patients diagnosed and treated in public hospitals of Mexico City during 2014–2016; the samples analyzed included 23 diagnosed acute lymphocytic leukemia (ALL) and 4 no-leukemia patients. The diagnosis of ALL was based on the histochemical tests and cytometric evaluation (monoclonal antibodies directed against lineage-associated antigens) of the bone marrow. Informed consent was obtained from each child′s parents. Clinical details and follow-up information were obtained from the medical records. A database was constructed to register the age, sex, residence, year of diagnosis, and clinical manifestations of the patients. The follow-up information during treatment was obtained in ALL patients.

### 4.2. RNA-Seq Libraries and Sequencing

RNA was extracted from mononuclear cell suspensions from diagnostic BM aspirates using the direct-zol RNA, and on-column DNase digestion was performed to remove DNA (Zymo Research, Irvine, CA, USA); the RNA integrity was assessed by using the 4200 TapeStation (Agilent Technologies, Santa Clara, CA, USA). The libraries were constructed using TruSeq Stranded Total RNA with Ribo-Zero Gold (Illumina, San Diego, CA, USA) according to the manufacturer’s instructions. Briefly, rRNA was removed from RNA samples and fragmented, and reverse transcriptase was used to synthesize cDNA and sequencing adapters were ligated. Libraries were evaluated using the 4200 TapeStation and sequenced for 2 × 75 cycles (paired-end sequencing) on the Illumina sequencing platform (NextSeq500, Illumina, San Diego, CA, USA). Raw reads were preprocessed using the standard Illumina pipeline which consists of dataset quality control using FastQC v0.11.5 (http://www.bioinformatics.babraham.ac.uk/projects/fastqc/; accessed on January 2018) for trimming and adapter removal Trimmomatic-0.36 (ww.usadellab.org/cms/index.php?page=trimmomatic; accessed on January 2018). TopHat (https://ccb.jhu.edu/software/tophat/index.shtml; accessed on January 2018) was used to align reads to an hg19 human genome reference and TopHat-fusion for gene fusion analysis.

### 4.3. Reverse Transcription–Polymerase Chain Reaction (RT-PCR) for Fusion Genes

Total RNA previously isolated from the bone marrow of patients was used for the synthesis of the cDNA using Superscript III Transcriptase (Invitrogen, Waltham, MA, USA), in accordance with the manufacturer’s instructions. Mononuclear cell suspensions previously obtained were subject to digestion with Proteinase K and subsequent phenol-chloroform was used for the extraction of genomic DNA. To detect ALL fusion, we performed PCR or RT-PCR using specific primers designed with Primer3Plus software (v. 0.4.0) [31], (Duke-NUS Medical School, Singapore) listed in Appendix A. We performed PCR with *Taq* DNA Polymerase (Invitrogene). The PCR products were electrophoresed in 2% agarose gel and cloned into vector pJET1.2 with CloneJET PCR Cloning Kit (Thermo Fisher Scientific, Waltham, MA, USA). The plasmids were confirmed by Sanger using an ABI3500xL genetic analyzer (Applied Biosystems, Waltham, MA, USA). Nucleotide sequences of each fusion gene have been deposited in GenBank under the accession numbers: MK172836, MK172837, MK172838, MK172839, MK172840.

## 5. Conclusions

The findings of the present study show that the identified novel fusion genes could be associated with the pathogenesis of childhood leukemia. Additionally, the relatively low frequency (39.13%) of translocations found in this subgroup of Mexican children with ALL confirms the high genetic variation of this disease. It would be important to identify the leading oncogenic lesions related to the evolution of the disease. In this work, we advanced the characterization of important genetic variations potentially related to the prognosis of ALL in Mexican children. In addition, the new fusion genes could contribute to better risk stratification and treatment.

## Figures and Tables

**Figure 1 ijms-20-02394-f001:**
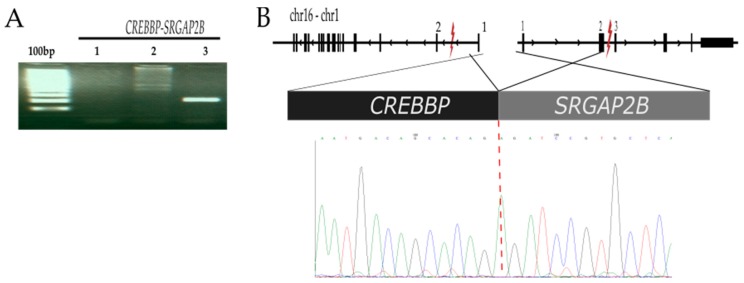
Characterization of the *CREBBP-SRGAP2B* gene fusion. (**A**) 100 bp marker, PCR of *CREBBP-SRGAP2B* using H_2_0 as negative control (lane1), DNA genomic (lane 2), and cDNA from total RNA isolated from 74MO sample (lane 3). (**B**) Schematic diagram showing the structure of *CREBBP-SRGAP2B* fusion, exon 1 of the *CREBBP* gene is fused in frame with exon 3 of the *SRGAP2B* gene. The symbol “lightning” indicates the breakpoint region. A *CREBBP-SRGAP2B* fusion sequence is shown below, with a schematic derived by the cloning and sequencing of the RT-PCR product.

**Figure 2 ijms-20-02394-f002:**
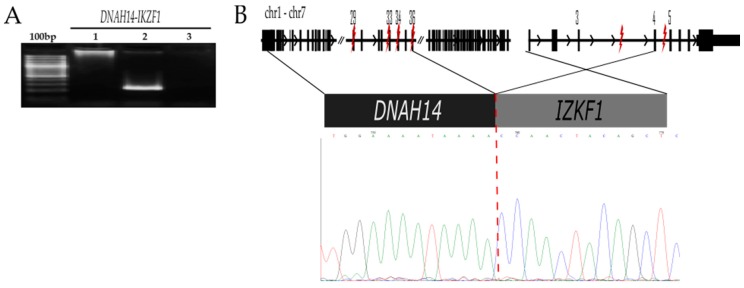
Detection of the *DNAH14-IKZF1* fusion sequence A. Genomic PCR products were obtained from original genomic DNA samples. We used primer sets at the intronic breakpoint between exon 36 of *DNAH14* with exon 4 of *IKZF1* (Appendix A). (**A**) 100 bp marker, PCR of *DNAH14-IKZF1* with genomic DNA of *DNAH14-IKZF1*-negative sample 74MO (lane 1), genomic DNA of 179 sample (lane 2), and H_2_O as negative control (lane 3). (**B**) Schematic representation of the *DNAH14-IKZF1* fusion. Exon 29, 33, and 34 of the *DNAH14* gene and exon 3 and 4 of the *IKZF1* gene could be fused together, synthesizing different fusion transcripts. The symbol “lightning” indicates the breakpoint region. Note: Only the intronic breakpoint between exon 36 of *DNAH14* with exon 4 of *IKZF1* was confirmed by cloning the sequencing of the PCR product.

**Figure 3 ijms-20-02394-f003:**
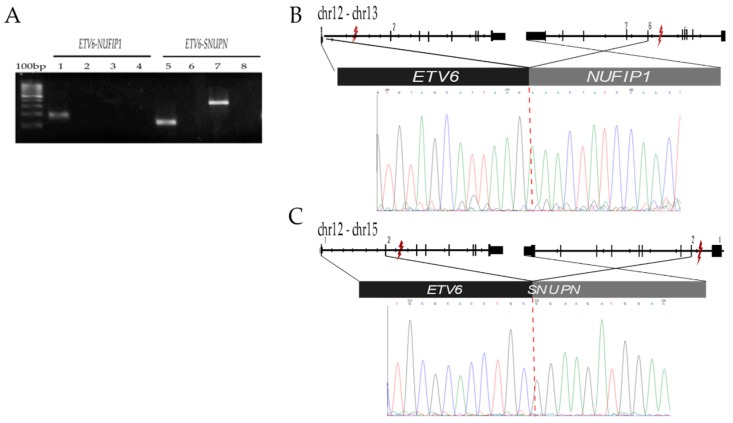
Identification of the *ETV6 gene* fusion transcript. For *ETV6* fusion confirmation, the original RNA sample was used. Primers were designed to amplify *ETV6*-*NUFIP1* with the two regions of 118 bp and 269 bp named internal and external, respectively. For *ETV6*-*SNUPN,* two regions, 112 bp-internal and 339 bp-external, were also amplified (Appendix A). (**A**) 100 bp marker, RT-PCR of *ETV6*-*NUFIP1* with primers ETV6-NUFIP1-int (lane 1), with primers ETV6-NUFIP1-ext (lane 3). RT-PCR with primers ETV6-SNUPN-int (lane 5), ETV6-SNUPN-ext (lane 7), and negative control (H_2_O) (lanes 2, 4, 6, and 8). (**B**) Schematic representation of the localization of the breakpoints within *ETV6* and *NUFIP1* and the corresponding result of sequencing where exon 1 of the ETV6 gene and exon 6 of the *NUFIP1* gene are fused together in this transcript. (**C**) Schematic representation of *ETV6-SNUPN* where exon 2 of the *ETV6* gene and exon 2 of the *SNUPN* gene are fused. The symbol “lightning” indicates the breakpoint region.

**Figure 4 ijms-20-02394-f004:**
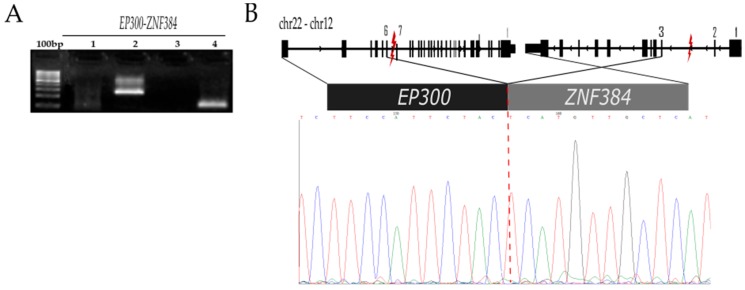
Identification of the *EP300-ZNF384* fusion. For the confirmation of *EP300-ZNF384* fusion, the original RNA sample was used. Primers were designed to amplify two regions of 106 bp and 250 bp, named internal and external, respectively (Appendix A). (**A**) 100 bp marker, RT-PCR of *EP300-ZNF384* with primers EP300-ZNF384-ext (lane 2), with primers EP300-ZNF384-int (lane 4), and negative control (H_2_O) (lanes 1 and 3). (**B**) Schematic representation of the localization of the breakpoints within *ZNF384* and *EP300* and the corresponding result of sequencing where exon 6 of the *EP300* gene and exon 3 of the *ZNF384* gene are fused in this transcript. The symbol “lightning” indicates the breakpoint region.

**Table 1 ijms-20-02394-t001:** Clinical characteristics and fusion gene present in patients with acute lymphoblastic leukemia.

Case	Age (Years)	Sex	Fusion Transcripts by NGS	Relapse	Death	Adherence	Diagnosis/Year	Initial WBC Count × 10^6^ Cell/L	BM Blast % at Diagnosis	IP	Outcome
74MO	10.8	M	*BCR-ABL* minor *SRGAP2B-CREBBP*	Yes	SS	Yes	ALL/2013	15,630	100	Pre-B	Progressed quickly; expired in 2 weeks
77MO	17.4	M	NF	No	No	Yes	ALL/2014	12,570	98	Pre-B	CR
197MO	9.4	F	*EP300-ZNF384*	No	No	Yes	ALL/2014	2430	95	Pre-B	CR
199MO	3.9	M	NF	No	SS	Yes	ALL/2014	440	100	Pre-B	Died after remission
63MO	2.0	M	*TCF3-PBX1*	No	SS	Yes	ALL/2015	7500	97	T cell	Died after remission
123MO	4.7	F	NF	No	No	Yes	ALL/2015	42,100	25	Pre-B	CR
269MO	4.0	F	*ETV6-RUNX1*	No	No	Yes	ALL/2015	9200	98	Pre-B	CR
273MO	14.8	M	NF	No	No	Yes	ALL/2015	12,460	96	Pre-B	CR
289MO	0.6	F	*MLL-AF4, GLYR1-SLC9A8* ^a^	No	No	Yes	ALL/2015	371,000	80	Pre-B	CR
374MO	4.4	F	NF	No	No	Yes	ALL/2015	20,220	100	Pre-B	CR
385MO	5.5	M	NF	ND	ND	A	ALL/2015	7200	85	Pre-B	ND
405MO	4.6	M	NF	No	No	Yes	ALL/2015	2700	25	Pre-B	CR
420MO	6.0	F	NF	No	No	Yes	ALL/2015	2360	90	Pre-B	CR
545MO	7.2	F	*WDR74-RCC1* ^a^	No	No	Yes	ALL/2015	8600	98	Pre-B	CR
549MO	4.9	M	NF	No	No	Yes	ALL/2015	13,300	98	Pre-B	CR
99MO	9.8	F	NF	No	No	Yes	ALL/2016	9000	90	Pre-B	CR
109MO	4.7	F	*NF*	No	No	Yes	ALL/2016	19,900	96	Pre-B	CR
122MO	12.3	M	NF	No	No	Yes	ALL/2016	4700	96	Pre-B	CR
179MO	1.8	F	*DNAH14-IKZF1*	No	No	Yes	ALL/2016	32,780	100	Pre-B	CR
196MO	4.1	F	NF	No	No	Yes	ALL/2016	2780	25	Pre-B	CR
369MO	2.3	M	NF	No	No	Yes	ALL/2016	2710	100	Pre-B	CR
546MO	13.0	M	NF	No	No	Yes	ALL/2016	8000	100	Pre-B	CR
28MO	10.3	F	*ETV6-SNUPN, ETV6-NUFIP1*	No	MOF TC, PN, TC.	Yes	ALL/2016	46,300	99.5	Pre-B	Progressed quickly; poorly responded to therapy, died after 2 weeks
73MO *	37.3	F	*NF*	ND	ND	ND	HLH/2014	2200	-	NA	NA
159MO *	5.8	M	NF	ND	ND	ND	EBV/2015	3620	15	NA	NA
165MO *	2.2	M	NF	ND	ND	ND	EBV/2015	29,740	-	NA	NA
83MO *	5.8	F	NF	ND	ND	ND	BCP/2017	2390	-	NA	NA

Abbreviations: ALL: acute lymphoblastic leukemia; IP: immunophenotype; NF: not found; NA; does not apply; EBV: Epstein–Barr virus infection; BCP: bicytopenia; SS: septic shock; ND: nondetermined; HLH: hemophagocytic lymphohistiocytosis; A: abandonment; MOF: multiple organ failure; TC: toxicity; PN: pneumonia; CR: complete remission; * negative for ALL; ^a^ not validated.

**Table 2 ijms-20-02394-t002:** In-frame fusion genes found in the acute lymphoblastic leukemia patients by RNA sequencing, genomic localization, and the associated phenotype.

Fusion	Gene Symbol (Chromosome Band)	Nucleotides (hg19)	Gene Description	Sample	Gene Previously Reported as Potential Prognostic Indicator	In-Frame	Fusion Validated
*CREBBP-SRGAP2B*	*SRGAP2B* (1q21.1)	144013900	SLIT-ROBO Rho GTPase-activating protein 2B	74MO	No reported	Yes	Yes
	*CREBBP* (16p13.3)	3929832	CREB-binding protein (CBP)		Mutations may confer to chemotherapy resistance and possibility of relapse [15]		
*DNAH14-IKZF1*	*DNAH14* (1q41.12)	225333860, 225333863, 225347499, 225354984, 225374260, 225346497	Dynein Axonemal Heavy Chain 14	179MO	No reported	Yes	Yes
	*IKZF1* (7p12.2)	50444490, 50367352, 50435762, 50444490, 50448363, 50444230	IKAROS Family Zinc Finger 1		Deletions and mutation were related to adverse prognosis, treatment failure, and risk of relapse [16,17,18]		
*ETV6-SNUPN*	*ETV6* (12p13.2)	11905512	ETS family transcription factor, Variant 6	28MO	In fusion with *RUNX1*, the most common genetic aberration in pediatric ALL and is related to favorable prognosis [19]	Yes	Yes
	*SNUPN* (15q24.2)	75913396	Snurportin 1		No reported		
*ETV6-NUFIP1*	*ETV6* (12p13.2)	11803093	ETS family transcription factor, Variant 6	28MO		Yes	Yes
	*NUFIP1* (13q14.12)	45540070	Nuclear Fragile X Mental Retardation Protein Interacting Protein 1		No reported		
*EP300-ZNF384*	*ZNF384* (12p13.31)	6788687	Zinc finger protein 384	197MO	*EP300-ZNF384* fusion is associated with a B-cell precursor ALL in childhood (3–4%) with better favorable response to chemotherapy than patients with MLL translocations [14]	Yes	Yes
	*EP300* (22q13.2)	41527639	E1A binding protein p300				

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
