# Peer review of "Identification and Characterization of Novel Fusion Genes with Potential Clinical Applications in Mexican Children with Acute Lymphoblastic Leukemia"

_ijms, 2019, doi:10.3390/ijms20102394_

Reviewer 1 Report

In the current manuscript by Minerva Mata-Rocha et al, the authors found several new fusion genes that have not been reported in young ALL patients. This work could potentially lead to subsequently biological studies and targeting treatment investigations. However, more information needs to be provided:

1. Are these new fusion genes co-occurring or excluded with other known mutations within patient samples? 2. Whether these new fusion genes have prognostic values?

Author Response

We thank to the reviewers for all their valuable comments which undoubtedly will improve the scientific quality of our report. We have replied to each of these comments in word file

Point 1: In the current manuscript by Minerva Mata-Rocha et al, the authors found several new fusion genes that have not been reported in young ALL patients. This work could potentially lead to subsequently biological studies and targeting treatment investigations. However, more information needs to be provided: 1. Are these new fusion genes co-occurring or excluded with other known mutations within patient samples?

 Response 1: Thank you for your comments. We agree with the fact that these fusion genes can co-occur with other known mutation because the genomic heterogeneity in ALL is large and each patient can have different mutations, deletions and fusion genes. However, we did not analyse if these fusion genes were co-ocurring with other known mutations because it was not the objective of the present research. This is an important limitation of our study, and it will be relevant to deeply explore this association in future studies. Therefore, we include this information as a study limitation in page 8 and 9, lines 38, and 1-4, respectively.

 Point 2: 2. Whether these new fusion genes have prognostic values?

Response 2: Thank you again for your valuable observation. We were not able to assess if these new fusion genes were associated with the prognosis of the disease because the small sample size. Noteworthy, it would be relevant to design and conduct a cohort study in order to elucidate the role of these novel fusion genes in the prognosis of ALL in Mexican children. 

Reviewer 2 Report

This is an interesting clinical observation that novel fusion genes were occurred in ALL patients. The authors used RNA-sequencing approach to identify four non-previously reported fusion genes, such as CREBBP-SRGAP2B, DNAH14-IKZF1, ETV6-SNUPN and ETV6-NUFIP1. Several suggestions list below may improve the scientific quality in the current manuscript. 1: I am wondering if karyotyping can identify the translocation of these fusion genes. The karyotyping of ALL patients would be better to show in figures. 2: For RNA-seq, the coverage and the reading depth is the important factor in analysis. This information is need to be displayed in either material and method section or result section. 3: The quality of tables is poor. It is barely to see the information of patients. 4: Supplementary materials can not be seem in this manuscript.

Author Response

We thank to the reviewers for all their valuable comments which undoubtedly will improve the scientific quality of our report. We have replied to each of these comments as follows

Point 1: This is an interesting clinical observation that novel fusion genes were occurred in ALL patients. The authors used RNA-sequencing approach to identify four non-previously reported fusion genes, such as CREBBP-SRGAP2B, DNAH14-IKZF1, ETV6-SNUPN and ETV6-NUFIP1. Several suggestions list below may improve the scientific quality in the current manuscript. 1: I am wondering if karyotyping can identify the translocation of these fusion genes. The karyotyping of ALL patients would be better to show in figures.

 Response 1: Thank you for your observations. Unfortunately, karyotyping of ALL patients in participating public hospitals of Mexico City is not routinely performed in ALL children because are hospitals with important limitations resource.

 Point 2: 2. For RNA-seq, the coverage and the reading depth is the important factor in analysis. This information is need to be displayed in either material and method section or result section

 Response 2: We agree. Therefore, we have included this information as supplementary figures (Supplementary Figure 1-5).

 Point 3: The quality of tables is poor. It is barely to see the information of patients

 Response 3: Thank you for your comments. For this reason, the resolution of the tables was improved in the modified version.

 Point 4: Supplementary materials cannot be seen in this manuscript.

 Response 4: In accordance with your comments, we have generated a supplementary material file.